# Evaluation of attraction and spatial pattern analysis of world cultural and natural heritage tourism resources in China

**Hui Zhang**[☯]**, Shujing Long**[iD]*[☯]

College of Tourism, Huaqiao University, Quanzhou, Fujian Province, China

☯ These authors contributed equally to this work.
* 1109136379@qq.com

## Abstract

The attraction of tourism resources is very important to promote the sustainable development of tourism industry. This study takes China's world cultural and natural heritage as the research object, and constructs an attractiveness evaluation system for China's world cultural and natural heritage tourism resources by collecting user feedback data from three major travel OTA platforms. At the same time, ArcGIS 10.7 software was used for spatial autocorrelation analysis and kernel density analysis to explore the spatial distribution pattern of tourism resource attraction. The results show that China's world cultural and natural heritage can be subdivided into 5 main categories and 10 sub-categories. From the perspective of spatial aggregation, only the Moran's $I$ index of tourist resource points showing a significant spatial aggregation feature. This study is helpful to reveal the weaknesses of tourism resource points and provide reference for sustainable development of attraction and optimization of tourism planning and management.

## Introduction

With the improvement of people's living standard, tourism has become the main choice of public entertainment. In the current tourism market, the rise of local tour, peripheral tour, micro vacation and other forms, as well as the emergence of new tourism products such as camping tour and light outdoor travel have further expanded the tourism market [1]. At the same time, the number of tourist destinations is increasing rapidly. Therefore, it is of great significance to study the attraction of tourist resources of tourist destinations to stand out in the highly competitive tourism market [2]. Tourism attraction is the basis of stimulating tourism activities and guiding tourists' travel intention and behavior. Tourism attractions have a macro impact on the overall development of tourism industry. However, tourism attraction should not only be evaluated from the passenger flow on demand side or the infrastructure on supply side. Tourism attraction is potential and variable, and not only depends on the supply of tourism resources itself, but also influenced by tourists' demand orientation for tourism resources [3]. Without attraction, there will be no tourism. Tourism attraction is the result of the combination of the internal "push" of tourists' travel intention and the external "pull" of

**Data Availability Statement:** All relevant data are within the manuscript and its Supporting Information files.

**Funding:** This work was supported by the Fujian Social Science Foundation Project Research on the

Driving Mechanism and Promotion Path of High-Quality Tourism Development in Fujian Province (Grant No. FJ2021B143).

**Competing interests:** The authors have declared that no competing interests exist.

tourism conditions [4]. Therefore, how to effectively transform tourism resources into tourism attractions and bring long-term development for scenic spots is a problem worth further discussion.

In tourism attraction, tourism resource attraction is particularly important. Foreign scholars define "tourism attraction" as "Tourist attraction", but nowadays scholars more often translate it as "tourism attraction", which is more similar to the concept of tourism resources in China [5]. Compared with tourism resources, the concept of tourism attractions is broader [6]. When superior tourism resources are fully utilized and become tourism attractions, it will become the iconic image of the tourist destination, improve tourism awareness, and bring huge passenger flow to it. At present, the research on tourism resource attraction mainly focuses on the evaluation of tourism resource attraction from the perspectives of IPA analysis [7,8], AHP analysis [9], tourist perception perspective [10,11], network information [12], etc., constructing tourism resource attraction model [13], and exploring its spatial structure characteristics [14]. Proposed the path to enhance the attraction of tourism resources [15]. However, at present, most relevant researches focus on the attractiveness of tourism resources in a certain province, city, or scenic spot in a certain case, and there are relatively few researches on a certain type of tourism attraction [16].

Compared with the popular Internet celebrity scenic spots, the world cultural and natural heritage sites with magnificent and beautiful natural scenery, profound and simple historical deposits and unique natural landscape are more likely to achieve long-term development and "long-term popularity". These tourism resources themselves have multiple values. It is worth paying attention to the sustainable development of heritage tourism to explore the attraction of world cultural and natural heritage tourism resources, explore their potential advantages and highlight them, and transform favorable influencing factors into the driving force for the sustainable development of tourist attractions [17]. With the maturity of mobile Internet technology and the popularization of 5G, the travel OTA platform is developing rapidly. Its functions cover six elements of tourism "food, accommodation, travel, shopping and entertainment", bringing one-stop service experience for tourists. Tourists can easily and quickly learn all the information about the destination through the travel platform before traveling. After the trip, they can post what they see, hear and feel through the tourism platform to provide other potential tourists with meaningful travel notes or tourism strategies [12]. Online OTA platform has become a necessary artifact for tourists to travel. Scholars have also begun to pay attention to the study of online data on tourism attraction. Online information data is characterized by large sample size, easy access and wide coverage [18]. The real feedback of user data from the tourism OTA platform is conducive to providing new research ideas and approaches for the attractiveness evaluation of tourism resources.

Therefore, this paper takes China's world cultural and natural heritage as the research object, uses three popular travel OTA platforms, Ctrip, Mafengwo and Qunar, as the data source, divides tourism resources into categories according to GB/T18972-2017 "Classification, Survey and Evaluation of Tourism Resources" national standard, and obtains relevant index data. To construct the evaluation system of attraction of China's world cultural and natural heritage tourism resources, and explore the characteristics of attraction of tourism resources. At the same time, ArcGIS 10.7 software was used to conduct spatial autocorrelation analysis and kernel density analysis, so as to understand the spatial distribution characteristics of attraction of China's world cultural and natural heritage tourism resources, and provide effective suggestions for the sustainable development of China's world cultural and natural heritage tourism resources and the optimization of tourism planning and management.

## Research samples and data sources

### Overview of the research sample

Due to its uniqueness, rarity and magnificence, world cultural and natural heritage has multiple values of aesthetics, literature, history and science, and has been included in the heritage protection list as a tourism attraction with relatively high development value [19]. Its own diversified value and image popularity have attracted much attention, but as a tourism attraction, in-depth exploration of its tourism resource attraction is conducive to determining the direction of the follow-up development of heritage resources and giving full play to the maximum tourism value and protection value [20]. By 2022, the total number of World heritage sites has reached 1,154, covering 167 countries in the world, and China has 56 world heritage sites in total [21]. Among them, China Danxia, the Karsts in southern China, royal tombs of Ming and Qing Dynasties, Quanzhou: the World maritime Trade Center of Song and Yuan Dynasties of China and other tourist resources are heritage tourism series, covering a number of scenic spots. In this study, 87 world cultural and natural heritage scenic spots in China were collected and sorted out as research samples, and those tourist resource spots that could not be accurately located, such as the habitat of migratory birds in the Yellow Sea (Bohai Sea) of China, were excluded.

### Data sources

This paper determines the number of Chinese cultural and natural heritages and the names of scenic spots according to the list of Chinese cultural and natural heritages published by the State Administration of Cultural Heritage [22]. The top OTA platforms in China, Ctrip, Mafengwo Tourism Official Website and Qunar, were used as data collection platforms to collect data of 21 evaluation indicators from eight dimensions of world cultural and natural heritage scenic spots. Among them, the dimensions of resource recreation value, tourist satisfaction, resource abundance, tourist sharing desire, resource popularity, resource reputation, resource development conditions and so on are derived from OTA platform user feedback data. Accessibility data came from Baidu Map and Gaode Map. The data was collected by the State Department of Natural Resources through December 31, 2022.

## Research design

### Classification of world cultural and natural Heritage tourism resources in China

Tourism resources cover a wide range, with diversity and overlap, so the academic community has diversified views on the classification of resource types [23–25]. In order to standardize the classification of tourism resources, China has published the national standard GB/T18972-2017 Classification, Investigation and Evaluation of Tourism Resources [16]. According to this standard, combining with the characteristics of the scenic spots in the World cultural and natural heritage list, this paper classifies the tourism resources of China's world cultural and natural heritage. According to the specific causes and status of existence, this paper divides China's world cultural and natural heritage tourism resources into five main categories: Territory cultural landscape, water landscape, biological landscape, architecture and facilities, and historical sites, and subdivides them into ten sub-categories. The detailed division of scenic resources can be seen in Table 1.

### Evaluation index system of tourism resource attraction

This paper evaluates tourism resources from the aspects of resource element value, resource influence and added value by referring to the scoring rules of the tourism resources evaluation

**Table 1. Classification system of World cultural and natural heritage tourism resources in China.**

| Main categories | Sub-categories | Fundamental type | Representative scenic spot |
|---|---|---|---|
| A.Territory cultural landscape (23) | AA.Natural landscape complex (10) | AAA.Hill-type landscape | Jiuzhaigou Scenic Area, Huanglong Scenic Area, Wulingyuan Scenic Area, Sanqingshan National Park, Xinjiang Tianshan Mountain, Honghe Hani Rice Terraces Cultural Landscape, Taishan, Huangshan, Wuyi Mountain, Hubei Shennongjia |
| | | AAB.Platform topographic landscape | |
| | | AAC.Gully landscape | |
| | | AAD.Beach topographic landscape | |
| | AB.Geological and structural features (1) | ABA.Biofossil site | Chengjiang Fossil Site |
| | AC.Surface configuration (12) | ACA.Terraced landscape | China Danxia (Danxia Mountain in Guangdong, Longhu Mountain in Jiangxi, Jianglang Mountain in Zhejiang, Chishui Danxia Tourism Area in Guizhou, Taining Scenic Tourism Area in Fujian, Langshan Mountain Scenic Area in Hunan), Karst in southern China (Kunming Shilin Natural Scenic Area, Libo Xiaoqi Hole in Guizhou, Wulong Xiannu Mountain National Forest Park in Chongqing, Shibing Yuntai Mountain in Guizhou, and Jinfo Mountain in Chongqing), Fanjing Mountain |
| | | ACB.Peak-pillar landscape | |
| | | ACC.Ridge landscape | |
| | | ACD.Gullies and caves | |
| | | ACE.Strange and pictographic rocks | |
| | | ACF.Geospheric disaster remains | |
| B.Water landscape (2) | BA.Lakes and marshes (1) | BBA.Recreational lake district | West Lake Cultural Landscape of Hangzhou |
| | | BAB.Ponds and pools | |
| | | BAC.Wetland | |
| | BB.Sea reef (1) | BEA.Recreational sea area | Gulangyu Island |
| | | BBB.Tidal surge and surf phenomenon | |
| | | BBC.Small island | |
| C.Biological landscape (5) | CA.Wildlife habitat (5) | CAA.Aquatic habitat | Giant panda habitat in Sichuan (Siguniang Mountain, Jiajin Mountain, Wolong Reserve in Sichuan), Hoh Xil in Qinghai, and migratory bird habitat in the Yellow Sea of China |
| | | CAB.Land animal habitat | |
| | | CAC.Bird habitat | |
| | | CAD.Habitat for butterfly | |
| D.Architecture and facilities (40) | DA.Cultural landscape complex (20) | DAA.Social and commercial activities | Great Wall, Mogao Grottoes, Temple of Heaven: Beijing Royal Altar, Mount Wutai, Beijing-Hangzhou Grand Canal, Quanzhou: The world maritime trade center in Song and Yuan Dynasties of China (Jiuri Mountain Wind-Praying Inscriptions, The Site of Maritime Trade Office, Ruins of Deji Gate, Tianhou Palace, Zhenwu Temple, Quanzhou Temple, Kaiyuan Temple, Statues of Laojun Rock, Qingjing Temple, Holy Tomb of Islam, Statue of Mani in Cao'an Temple, Luoyang Bridge, Anping Bridge, Liusheng Tower, Wanshou Tower) |
| | | DAB.Military sites and ancient battlefields | |
| | | DAG.Places of worship and sacrifice | |
| | | DAH.Transportation yard station | |
| | | DAI.Memorial sites and places of commemoration | |
| | DB.Functional buildings and core facilities (18) | DBA.Characteristic urban block | Mausoleum of Emperor Qin Shihuang and Pit of Terracotta Warriors and horses, Qingcheng Mountain and Dujiangyan, ancient villages in south Anhui—Xidi, Hong Village, royal tombs of Ming and Qing Dynasties (Obvious Mausoleum, East Tomb of Qing Dynasty, West Tomb of Qing Dynasty, Xiaoling Tomb of Ming Dynasty, Thirteen Tombs of Ming Dynasty, Fulling Tomb of Qing Dynasty, Zhaoling Tomb of Qing Dynasty), Imperial City of Koguryeo, Historical City of Macao, Longmen Grottoes, Yungang Grottoes, Fujian Tulou (Yongding Tulou, Zhangzhou Nanjing Tulou), Kaiping Diaolou and villages |
| | | DBB.Characteristic house | |
| | | DBC.An independent hall, room, pavilion | |
| | | DBD.Independent field, office | |
| | | DBE.Cavern | |
| | | DBF.Mausoleum | |
| | DC.Landscape and sketch architecture (2) | DCA.Image marker | Suzhou classical Garden, the Summer Palace |
| | | DCB.Scenic lookout | |
| | | DCC.airy pavilions and pagodas | |

(*Continued*)

**Table 1.** (Continued)

| Main categories | Sub-categories | Fundamental type | Representative scenic spot |
|---|---|---|---|
| E.Historical sites (17) | EA.Material cultural relics (17) | EAA.Architectural remains | Forbidden City (Beijing Forbidden City, Shenyang Ancient Palace), Chengde Summer Resort and its surrounding temples, Qufu Sankong Scenic Spot, Wudang Mountain Ancient Architecture Complex, Lhasa Potala Palace Historical Complex, Lushan Mountain National Park, Lijiang Ancient City, Pingyao Ancient City, Dazu Stone Carvings, Yin Ruins, Yuan Shangdu Site, Tangya Tusi Site, Zuojiang Huashan Rock Painting Art and Cultural Landscape, Liangzhu Ancient City Site, Mount Emei, Leshan Giant Buddha |
| | | EAB.Movable relic | |

part of the national standard GB/T18972-2017 "Tourism Resources Classification, Investigation and Evaluation". Based on previous studies [26], this paper mainly constructs an evaluation system for the attraction of China's world cultural and natural heritage tourism resources from eight dimensions, including "resource recreation value", "tourist satisfaction", "resource abundance", "tourist sharing desire", "resource popularity", "resource reputation", "resource development conditions", and "accessibility" (as shown in Table 2). These eight dimensions run through the evaluation of the types of resources perceived by tourists, tour value, experience value, sharing value and tourism transportation during the whole tour process, which can measure the attraction of tourism resources from the perspective of tourists in a more comprehensive way.

In order to ensure the authenticity and availability of the data, the data in this paper are from the real ratings of tourists on the travel OTA platform. The dimension of "resource recreation value" comes from Ctrip, including four aspects of "scenery score", "fun score", "cost performance score" and "star rating of scenic spot" to evaluate the appreciation, fun and cost performance of tourism resources [27]. From the dimension of "tourist satisfaction", the overall scores of world cultural and natural heritage scenic spots were collected from the three travel OTA platforms of Mafengwo Tourism Official Website [28], Ctrip [27] and Qunar [29] to measure the degree to which tourist destinations meet the needs of tourists. "Resource abundance" refers to the concentration degree of tourism attractions in heritage tourism to a certain extent. This paper chooses the number of pictures in online comments to measure resource abundance. Recommendable scenic spots indicate that tourism resources are superior and attractive. Therefore, "tourist sharing desire" is an indicator to measure tourists' fondness for tourism resources and the degree of dissemination and recommendation, which is mainly represented by the number of travel articles and notes. "Resource popularity" refers to the degree to which tourism resources are understood by the outside world. The number of comments can reflect the degree of attention of tourists. "Resource reputation" is an index to evaluate tourists' recognition of tourism resources, measured by the proportion of the number of favorable comments to the total number of comments. "Resource development conditions" reflect the ticket income of scenic spots, and to some extent reflect the consumption of tourists in scenic spots; "Accessibility" is an index to measure the accessibility of traffic in the scenic area, which mainly considers the distance between the city center and the scenic area and the airport and the scenic area. It is worth noting that "resource development conditions" and "accessibility" are negative indicators, while the rest are positive indicators.

In this paper, the entropy method is selected to determine the weight of each index of the attraction of world cultural and natural heritage tourism resources. The greater the information entropy, the higher the degree of order, and vice versa. The entropy method in the assignment method avoids the arbitrariness of subjective weighting, with strong rigor and objectivity, and high accuracy [30]. The specific calculation steps of entropy method are shown as follows:

## (1) Dimensionless processing

Since variables involve different units, their meanings and dimensions of measurement will be different, which cannot be calculated and compared directly. Therefore, it is necessary to standardize the original data to ensure that the data will be interfered by other factors to a minimum extent. Therefore, in this paper, the collected network data will be dimensionless processed according to positive and negative indicators. The formula is as follows:

$$Z_{ij} = \frac{X_{ij} - X_{min}}{X_{max} - X_{min}} *100 \quad \text{(Positive index)} \tag{1}$$

$$Z_{ij} = \frac{X_{max} - X_{ij}}{X_{max} - X_{min}} *100 \quad \text{(Negative index)} \tag{2}$$

**Table 2. Evaluation index system of attraction of China's World cultural and natural heritage tourism resources based on network information.**

| First-order index | Secondary index | Indicator source |
|---|---|---|
| Resource recreation value (2.54%) | Scenery score (18.29%) | Trip.com Group |
| | Fun score (19.11%) | |
| | Cost performance score (16.4%) | |
| | Star rating of scenic spot (46.2%) | |
| Tourist satisfaction (2.05%) | Overall score (37.37%) | Mafengwo Tourism Official Website |
| | Overall score (31.44%) | Trip.com Group |
| | Overall score (31.19%) | Qunar Cayman Islands Limited |
| Resource abundance (16.48%) | Picture (48.34%) | Mafengwo Tourism Official Website |
| | Picture (51.66%) | Trip.com Group |
| Tourist sharing desire (27.55%) | Travel notes (35.31%) | Mafengwo Tourism Official Website |
| | Travel notes (30.84%) | Trip.com Group |
| | Notes (33.85%) | Qunar Cayman Islands Limited |
| Resource popularity (30.72%) | Comment number (26.93%) | Mafengwo Tourism Official Website |
| | Comment number (32.12%) | Trip.com Group |
| | Comment number (40.96%) | Qunar Cayman Islands Limited |
| Resource reputation (19.51%) | Favorable rate (1.3%) | Mafengwo Tourism Official Website |
| | Favorable rate (56.83%) | Trip.com Group |
| | Favorable rate (41.86%) | Qunar Cayman Islands Limited |
| Resource development conditions (0.53%) | Ticket price (¥) | Trip.com Group |
| Accessibility (0.63%) | The distance between the scenic area and the nearest airport (KM) (72.04%) | Baidu Map |
| | The distance between the scenic spot and the city center (KM) (27.96%) | Amap |

All relevant data are within the manuscript and its S1 Data.

### (2) Normalization treatment

Calculate the proportion of the j-th index in this index.

$$P_{ij} = \frac{Z_{ij}}{\sum_{i=1}^{m} Z_{ij}} \tag{3}$$

### (3) Calculate index entropy

$$E_j = -K \sum_{i=1}^{m} P_{ij} ln P_{ij} \tag{4}$$

### (4) Calculate the redundancy of information entropy

$$D_j = 1 - E_j \tag{5}$$

### (5)Calculate index weight

$$W_j = \frac{D_j}{\sum_{j=1}^{n} D_i} \tag{6}$$

### (6)Calculate the composite index of each secondary index and each dimension

$$C_i = Z_{ij} \times W_j \tag{7}$$

### Exploratory Spatial Data Analysis (ESDA)

The attraction of tourism resources is affected by the mutual influence of adjacent resources, and there is a certain correlation. Based on the spatial global autocorrelation in exploratory spatial analysis (ESDA), this paper presents the analysis results by Moran's $I$ index and ArcGIS software to explore the spatial dispersion and aggregation of tourism resources. The calculation formula is as follows:

$$I = \frac{n\sum_{i=1}^{n}\sum_{j=1}^{n} W_{ij} \ (X_i - \bar{X}) \ (X_j - \bar{X})}{\sum_{i=1}^{n}\sum_{j=1}^{n} W_{ij} \ (X_i - \bar{X})^2} \tag{8}$$

Generally, when the value is greater than zero, the positive correlation is presented and the spatial distribution is aggregated. When the value is zero, the space has no correlation; When the value is less than zero, the correlation is negative and the spatial distribution is discrete [31].

### Nuclear density analysis

Kernel density analysis is an index that reflects the influence intensity of factors on the surrounding area [32]. The specific formula of kernel density is as follows:

$$f(x) = \frac{1}{nh} \sum_{i=1}^{n} k\left(\frac{x - x_i}{h}\right) \tag{9}$$

$k\left(\frac{x-x_i}{h}\right)$ is the kernel function; h is the radius; n is the number of world cultural and natural heritage sites in China; $(X - X_i)$ is the distance from the estimated point X to the sampling point. The greater the value of $f(x)$, the stronger the attraction of heritage tourism resources and the better the development [33].

# Evaluation on the attractiveness of China's World cultural and natural heritage tourism resources

## Overall tourism resource attraction evaluation

With reference to the World Cultural and Natural Heritage List of China, a total of 87 tourism resource sites were evaluated, including 23 geographical and cultural landscapes, 2 water landscapes, 5 biological landscapes, 40 architectural and facility landscapes and 17 historical sites, covering 10 sub-categories of tourism resource sites (see Table 3 for details). The overall attraction index of world cultural and natural heritage tourism resource points is 0.1511. Among them, the number of cultural landscape complex is the largest, followed by practical buildings and core facilities and material cultural relics. From the main attraction index, the tourism attraction index of Class A geological landscape is 0.088, that of class B water landscape is 0.337, that of class C biological landscape is 0.053, that of class D buildings and facilities is 0.121, and that of class E historical sites is 0.154. Therefore, the tourism resources of water landscape are the most attractive, indicating that for tourists, the landscape near the water can stimulate tourists' travel intention more. The biological landscape is the least attractive, because the biological protection area needs to pay attention to the protection of resources, and the tourism development is less. In terms of the attractiveness index of subcategories, the top five subcategories of the attractiveness index are BB sea reefs, BA limnology, DC landscape and sketch architecture, EA material cultural relics and AA natural landscape complex. From the attractiveness index of tourist resource points, the top five scenic spots are the Forbidden City in Beijing, the Old Town of Lijiang, the Tomb of Qin Shihuang and the Pit of Terracotta Warriors and Horses, Gulangyu Island and the West Lake cultural landscape in Hangzhou.

**Table 3. The attractiveness evaluation scores of various tourism resources of China's world cultural and natural heritage.**

| Main categories | Sub-categories | Attractiveness index | Resource recreation value | Tourist satisfaction | Resource abundance | Tourist sharing desire | Resource popularity | Resource reputation | Resource development conditions | Accessibility |
|---|---|---|---|---|---|---|---|---|---|---|
| A.Territory cultural landscape | AA.Natural landscape complex | 0.141 | 0.864 | 0.829 | 0.121 | 0.157 | 0.076 | 0.038 | 0.568 | 0.797 |
| | AB.Geological and structural features | 0.040 | 0.603 | 0.425 | 0.004 | 0.004 | 0.001 | 0.021 | 0.926 | 0.793 |
| | AC.Surface configuration | 0.085 | 0.817 | 0.748 | 0.059 | 0.059 | 0.033 | 0.025 | 0.589 | 0.701 |
| B.Water landscape | BA.Lakes and marshes | 0.292 | 0.962 | 0.913 | 0.438 | 0.251 | 0.303 | 0.025 | 0.655 | 0.927 |
| | BB.Sea reef | 0.382 | 0.786 | 0.784 | 0.578 | 0.417 | 0.396 | 0.023 | 0.771 | 0.968 |
| C.Biological landscape | CA.Wildlife habitat | 0.053 | 0.592 | 0.710 | 0.018 | 0.021 | 0.009 | 0.025 | 0.836 | 0.493 |
| D. Architecture and facilities | DA.Cultural landscape complex | 0.077 | 0.579 | 0.611 | 0.052 | 0.041 | 0.037 | 0.041 | 0.857 | 0.928 |
| | DB.Functional buildings and core facilities | 0.116 | 0.804 | 0.789 | 0.103 | 0.069 | 0.076 | 0.055 | 0.694 | 0.781 |
| | DC.Landscape and sketch architecture | 0.172 | 0.931 | 0.841 | 0.169 | 0.126 | 0.173 | 0.025 | 0.936 | 0.906 |
| E.Historical sites | EA.Material cultural relics | 0.154 | 0.769 | 0.755 | 0.162 | 0.146 | 0.128 | 0.023 | 0.639 | 0.806 |

From the score of individual resources, it can be seen that the historical relics resources can better promote the tourism attraction due to their large volume and profound cultural deposits.

In addition, the top five scenic spots of landscape tourism resources are Huangshan Mountain, Jiuzhaigou Scenic spot, Libo Xiaoqikong Scenic spot in Guizhou, Taishan and Huanglong Scenic spots, which are mainly natural landscape complex. Gulangyu Island and Hangzhou West Lake Cultural Scenic spot were ranked in the list of water landscape tourism resources. The top five biological landscape tourism resources are Sigungwu Mountain in Sichuan Province, Hoh Xil National Nature Reserve in Qinghai Province, Wolong National Nature Reserve in Sichuan Province, migratory bird habitat in the Yellow Sea and Jiajinshan Mountain in Sichuan Province. The top five tourist resources are the Mausoleum of Emperor Qin Shihuang and Pit of Terracotta Warriors, the Summer Palace, Mogao Grottoes, Badaling Great Wall and Hongcun, an ancient village in southern Anhui, which are mainly cultural landscape complex. The top five historical sites are the Forbidden City in Beijing, the Old Town of Lijiang, the Potala Palace in Lhasa, the Ancient City of Pingyao and Mount Emei.

## Score evaluation of resource elements

The scores of basic types of tourism resource points in the secondary index were analyzed. In terms of resource recreation value, the top three tourist resource points were the Forbidden City, the Summer Palace and Huangshan Mountain. The top three subcategories are BA lakes and marshes, DC landscape and sketch architecture and AA natural landscape complex. Biological landscape and historical relics resource categories are less attractive, indicating that they are relatively boring or lack of scenery and low cost performance for tourists. In the future development, cultural and tourism linkage can increase the interest and experience of sightseeing.

In terms of tourist satisfaction, the top three tourist resources are the Forbidden City in Beijing, the Mausoleum of Emperor Qin Shihuang, the Pit of Terracotta Warriors and Mount Huangshan, all of which are scenic spots with profound tourism resources and culture, mature development and perfect tourism facilities and services, with high tourist satisfaction. The top three subcategories are BA lakes and marshes, DC landscape and sketch architecture and AA natural landscape complex, which are the same as the category of recreation value of resources, indicating that recreation value indirectly affects tourist satisfaction and is an important criterion for evaluating tourist satisfaction.

In terms of resource abundance, the top three tourist resources are the Forbidden City in Beijing, the Mausoleum of Qin Shihuang, the Pit of Terracotta Warriors and the Ancient Town of Lijiang, which is measured by the number of pictures on the tourism platform. The more photos, the better the overall picture of the scenic spot can be reflected and the interest of tourists can be inspired, thus becoming an important way to enhance the attraction of tourism resources. The top tourist resource points are all tourist attractions with high tourism visibility and must-punch in cities, which reflects the popular tourism image has a great impact on tourist attraction. The top three subcategories are BB sea reefs, BA lakes and marshes and DC landscape and sketch architecture, while AB geological and structural features, CA wildlife habitat and DA cultural landscape complex rank the bottom. The development and management of these types of tourism resources focus on protection, so it is particularly important to improve China tourism quality to meet the higher tourism needs of tourists.

In terms of tourist sharing desire, the top three tourist resources are the Old Town of Lijiang, the Tomb of Qin Shihuang, the pit of Terracotta Warriors and Horses, and the Mogao Grottoes, which reflects the strong uniqueness and time-out of tourism resources, and is an

important factor for tourists to spontaneously promote the tourism image. The top three sub-categories are BB sea reefs, BA lake marshes and AA natural landscape complex. It can be seen that terrain and water landscape are the first choice of tourists for leisure vacation, and they like to publish travel guides about terrain and water landscape on the Internet. In terms of expressing emotions, writing tourism strategies and promoting, they have good publicity and sharing effect.

In terms of resource popularity, the top three tourism resource spots are the Forbidden City in Beijing, the Tomb of Qin Shihuang, the Pit of Terracotta Warriors and Horses, and Gulangyu Island, which respectively represent the city of Beijing, Xi'an and Xiamen, and their popularity is deeply rooted in the hearts of the people. The top three subcategories are BB sea reefs, BA lake marshes, and DC landscape and sketch architecture, while AB geological and structural features, CA wildlife habitat, and AC surface configuration. These scenic spots are mostly protected by biology and geology, and the degree of tourism development and publicity is low, which is one of the important factors leading to the low tourism popularity.

In terms of resource reputation, the top three tourist resources are Jingling Mausoleum, Wutai Mountain and Hubei Shennongjia National Nature Reserve. Different from the scenic spots with the highest level of tourist satisfaction, it shows that tourist satisfaction does not necessarily lead to reputation. This kind of scenic spots can give tourists unexpected travel experience, thus improving their reputation. The top three subcategories are DB functional buildings and core facilities, DA cultural landscape complex, and AA natural landscape complex.

In terms of resource development conditions, Quanzhou's Song and Yuan China's World Marine Trade Center heritage scenic spots are many and simple, most of them do not set tickets, and many scenic spots are rated at the top of resource development conditions. The top three subcategories are DC landscape and sketch architecture, AB geological and structural features, DA cultural landscape complex; The AA natural landscape complex, which ranks the bottom, has a larger general scenic area and a higher admission fee, thus affecting its score in terms of resource development conditions.

In terms of accessibility, the top three tourist resources are Yin Ruins, Ming Tombs and Macao Historic City. Most of these large-scale scenic spots are located in the suburbs of cities, with similar site selection concepts to airports, so they are close to transportation hubs and have strong accessibility. The top three subcategories are BB sea reefs, DA cultural landscape complex and BA lake marshes. At the bottom of the list are Hoh Xil National Nature Reserve in Qinghai Province, Jiajin Mountain in Sichuan Province and Taining Scenic Tourist Area in Fujian Province. These scenic spots have inconvenient transportation and low accessibility.

## Analysis on spatial pattern of attraction of World cultural and natural Heritage tourism resources in China

### Global autocorrelation analysis of tourism resource attraction

In this paper, the attractiveness of all tourist resource points and A cultural landscape, B water landscape, C biological landscape, D buildings and facilities, E historical sites resource points divided by main categories is calculated by Moran's $I$ index, and the results are shown in Table 4. It is found that the value of attraction of all tourist resource points and building and facility resource points is less than 0, and the value of attraction of other tourist resource points is greater than 0, and $I_C < I_E < I_B < I_A$. The results of Moran's $I$ index show that the attraction of all tourism resource points and building and facility resource points is less than 0, showing discrete characteristics, but its P value is greater than 0.05, failing the significance test, indicating that the attraction of world cultural and natural heritage tourism resource points and building

**Table 4. Moran's *I* index of tourism resource attraction.**

| Type | Moran's *I* | Z value | P value |
|---|---|---|---|
| All tourist resource points | -0.0424 | -0.204 | 0.8383 |
| A.Territory cultural landscape | 0.21 | 3.061 | 0.0022 |
| B.Water landscape | 0.052 | 1.271 | 0.204 |
| C.Biological landscape | 0.028 | 1.2645 | 0.206 |
| D.Architecture and facilities | -0.0483 | -0.3777 | 0.7057 |
| E.Historical sites | 0.0337 | 0.7913 | 0.4288 |

and facility resource points are randomly distributed, without obvious discrete characteristics. This is mainly because, among the tourist resource sites listed as World cultural and natural heritage sites, buildings and facilities mostly show regional characteristics and are limited by geographical factors. Moreover, the large scale of development will not show the state of clustered development. The Moran's *I* of landscape A, water landscape B, biological landscape C and historical site E are all greater than 0, showing a positive spatial correlation, indicating a spatial agglomeration feature. However, from the point of significance P value, the P value of resource point attraction is less than 0.05, only landscape, while other tourism resource types did not pass the significance test of P value, there is no statistical significance. Therefore, only the Moran's *I* of landscape A is greater than 0, the P value is less than 0.01, and the Z score is greater than 2.58, indicating that tourism resources of landscape A present obvious agglomeration characteristics in spatial distribution.

## Kernel density analysis of tourism resource attraction

This paper conducts A kernel density analysis on the attractiveness of all tourism resource points and the resource points A, B, C, D, E. The attraction of all tourist resource points is centered in East China, Southwest China and North China and dispersed around, and the most attractive areas are coastal areas. The tourist attraction of territory cultural landscape is mainly concentrated in East China, Southwest China and Central China, especially in East China and Southwest China, where the density of attraction is the strongest. The water landscape in East China has the strongest attraction of tourism resources. They are all famous water scenic spots with strong attraction. The attraction of biological landscape tourism resources is mainly in southwest China, which is mainly because Yunnan, Sichuan and other southwestern provinces have superior geographical conditions and rich biodiversity. The attraction of tourism resources of architecture and facilities is mainly concentrated in East China and North China, among which East China is more attractive, mainly because these areas have unique architectural style and folk culture; The attraction of tourism resources of historical sites is mainly in southwest China and North China, and the attraction decreases outward successively, showing a strong trend of many points. In general, East China and Southwest China are endowed with unique natural resources, diversified resource types, and the attraction density remains stable. They have good basic tourism conditions to maintain and enhance the attraction of tourist resources in scenic spots.

## Conclusion and discussion

This paper takes China's world cultural and natural heritage as the research object, and classifies the resources according to GB/T18972-2017 "Classification, Survey and Evaluation of Tourism Resources". In terms of online information acquisition channels, this paper selects three popular travel OTA platforms, Ctrip, Mafengwo Tourism Official Website and Qunar, to

construct an evaluation index system for the attractiveness of China's world cultural and natural heritage tourism resources. ArcGIS 10.7 software was used for spatial autocorrelation analysis and kernel density analysis to understand the spatial distribution pattern of attraction of world cultural and natural heritage tourism resources in China. The study reached the following conclusions:

First, in this paper, 87 scenic spots involved in China's world cultural and natural heritage tourism resources are subdivided into 5 main categories and 10 sub-categories according to resource categories. In the main category, there are 2 tourist resource points of water landscape; Buildings and facilities had the largest number of tourist resources, with a total of 40, followed by cultural landscape (23), historical sites (17), and biological landscape (5). In the subcategory, the cultural landscape complex has the largest number of tourist resources (20), while the number of geological and structural features, lakes and marshes, and islands and reefs on the sea are relatively small, each with only 1, which reflects the long formation cycle, small number of individual and high implied value of naturally formed geological and water-shaped tourism resources to a certain extent.

Second, the average overall attractiveness of China's world cultural and natural heritage tourism resources is 0.1511. Among them, Category B water landscape tourism resource point has the highest tourism attraction index (0.337), followed by category E historical sites tourism resource point (0.154), category D architecture and facilities tourism resource point (0.121), category A territory cultural landscape tourism resource point (0.088), and Category C biological landscape tourism resource point (0.053). Among the subcategories, the attraction index of sea islands, lakes and marshes and landscape features ranks the top three, while the attractions of Beijing's Forbidden City, Lijiang Ancient City, Qin Shi Huang's Tomb and Terracotta Warriors pit rank the top three, and the tourism resources of historical sites account for a large proportion. At the bottom of the list are the Holy Sepulchre of Islam, Liu Sheng Tower and Deji Gate Site, all of which are cultural landscape complexes. This indicates that although there are more tourist resources in the cultural landscape complex, the development level of its perception to tourists is insufficient, which cannot drive the high-quality development of scenic tourism, resulting in a low attractiveness index.

Third, in this study, ArcGIS 10.7 software was used for spatial autocorrelation analysis and kernel density analysis to explore the spatial distribution characteristics of China's world cultural and natural heritage tourism resources. The results show that, from the perspective of spatial aggregation, only the tourist resource point Moran's $I$ is greater than 0 and P value is less than 0.05, showing a significant spatial aggregation feature, with East China, Southwest China and Central China as the core. All the attractions of tourist resource points are mainly in East, southwest and North China. The attraction of tourism resources of water landscape is concentrated in the coastal areas of East China. The southwest region is the gathering area of tourist resource attraction of biological landscape. The attraction of tourism resources of architecture and facilities is mainly concentrated in East and North China. The attraction of tourism resources of historical sites is mainly in southwest and North China.

This study reflects the attractiveness of China's world cultural and natural heritage in different dimensions of tourism resources and provides guidance and suggestions for future scenic spots to strengthen the construction of tourism attractions. China's world cultural and natural heritage is not balanced in spatial distribution, but also in terms of attraction, there are resources and regional disparities. In the future development planning and layout, the local governments of the world heritage tourism resource points should pay attention to the attractive highlights of the categories they belong to, start from the advantages, complement the weaknesses, and advance together, so that the world heritage sites can win the development advantages among the Internet celebrity scenic spots and achieve sustainable development. In

addition, this paper focuses on different types of China's world cultural and natural heritage tourism resources, discusses their spatial distribution characteristics, and provides suggestions for other types of A-level tourism resources to play A comparative advantage in space, which is helpful for rational planning and improving the transformation of local tourism resources to tourism attractions. However, this paper also has some shortcomings, such as not considering the volume of world cultural and natural heritage sites; The classification of world heritage resource points determined at different time points has not been studied. In the future, research samples can be further refined, and methods such as fsQCA can be used to explore ways to enhance the attractiveness of China's world cultural and natural heritage tourism resources, so as to promote the long-term development of China's precious cultural tourism resources.

## Supporting information

**S1 Data.**
(XLSX)

## Acknowledgments

Thanks to all the authors who contributed to this article and approved the submitted version.
Acknowledgments: Thanks to everyone who helped with the paper.

## Author Contributions

**Conceptualization:** Hui Zhang.

**Data curation:** Hui Zhang.

**Methodology:** Hui Zhang, Shujing Long.

**Resources:** Shujing Long.

**Software:** Shujing Long.

**Supervision:** Hui Zhang.

**Validation:** Hui Zhang, Shujing Long.

**Visualization:** Hui Zhang, Shujing Long.

**Writing – original draft:** Shujing Long.

**Writing – review & editing:** Shujing Long.

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
