## [Decision Letter · Decision Letter 0]

16 May 2023

PONE-D-23-13162Evaluation of attraction and spatial pattern analysis of world cultural and natural heritage tourism resources in ChinaPLOS ONE

Dear Dr. Long,

Thank you for submitting your manuscript to PLOS ONE. After careful consideration, we feel that it has merit but does not fully meet PLOS ONE’s publication criteria as it currently stands. Therefore, we invite you to submit a revised version of the manuscript that addresses the points raised during the review process.

We look forward to receiving your revised manuscript.

Kind regards,

Qingsong He, Ph.D.

Academic Editor

PLOS ONE

Journal Requirements:

3.  In your Methods section, please include additional information about your dataset and ensure that you have included a statement specifying whether the collection and analysis method complied with the terms and conditions for the source of the data.

4."In your Data Availability statement, you have not specified where the minimal data set underlying the results described in your manuscript can be found. PLOS defines a study's minimal data set as the underlying data used to reach the conclusions drawn in the manuscript and any additional data required to replicate the reported study findings in their entirety. All PLOS journals require that the minimal data set be made fully available. For more information about our data policy, please see http://journals.plos.org/plosone/s/data-availability.

We will update your Data Availability statement to reflect the information you provide in your cover letter."

6. We note that Figure (1-6) in your submission contain copyrighted images. All PLOS content is published under the Creative Commons Attribution License (CC BY 4.0), which means that the manuscript, images, and Supporting Information files will be freely available online, and any third party is permitted to access, download, copy, distribute, and use these materials in any way, even commercially, with proper attribution. For more information, see our copyright guidelines: http://journals.plos.org/plosone/s/licenses-and-copyright.

1. You may seek permission from the original copyright holder of Figure (1-6) to publish the content specifically under the CC BY 4.0 license. 

Reviewers' comments:

Reviewer's Responses to Questions

**Comments to the Author**

1. Is the manuscript technically sound, and do the data support the conclusions?

Reviewer #1: Yes

2. Has the statistical analysis been performed appropriately and rigorously? 

Reviewer #1: Yes

3. Have the authors made all data underlying the findings in their manuscript fully available?

Reviewer #1: Yes

4. Is the manuscript presented in an intelligible fashion and written in standard English?

Reviewer #1: Yes

5. Review Comments to the Author

Reviewer #1: This study intend to Evaluation of attraction and spatial pattern analysis of world cultural and natural heritage tourism resources in China. Below are some of my key observations:

(1) Methodology. The author(s) need to reconsider their research design. This paper is a technology oriented paper or a empirical paper. If it is a technology oriented paper, then they have to pay more attention to develop some new algorithm or methods such as to improve the accuracy of the prediction. On the other hand, it is an empirical paper, the author(s) need to have a conceptual framework (or some analytical models).

(2) The author(s) need to think about the implication of this paper. Can the results of this paper present some new insight to the scholars or can the results give some practical suggestions to the government?

6. PLOS authors have the option to publish the peer review history of their article (what does this mean?). If published, this will include your full peer review and any attached files.

Reviewer #1: No

---

## [Author Response · Author response to Decision Letter 0]

7 Jul 2023

Respected reviewers and editors:

Hello! First of all, I would like to thank the reviewers and editor for their valuable comments in their busy schedule. The reviewer and editor’s suggestions have provided very useful guidance for the revision and improvement of this paper.After receiving the evaluation of the article, I was very glad and nervous. I was glad that the teachers affirmed the topic selection of the article. At the same time, I also revised the article carefully. I consulted professors and teachers for many times, and combined with the suggestions of reviewers, I carefully revised and improved the paper as follows ( the blue font refers to the suggestions of reviewers, the black font refers to the revised reply instructions, and the red font refers to the detailed modifications in the paper).

1.Methodology. The author(s) need to reconsider their research design. This paper is a technology oriented paper or a empirical paper. If it is a technology oriented paper, then they have to pay more attention to develop some new algorithm or methods such as to improve the accuracy of the prediction. On the other hand, it is an empirical paper, the author(s) need to have a conceptual framework (or some analytical models).

Thank you very much. This paper belongs to the empirical research paper, compared with the technical paper, it is more inclined to use big data in resource evaluation to prove the attraction of tourism resources. According to the scoring rules of the tourism resources evaluation part of the Chinese national standard GB/T18972-2017 "Classification, Investigation and Evaluation of tourism resources", this paper evaluates tourism resources from the aspects of resource element value, resource impact and added value. The evaluation index system is constructed to put forward a set of operable index sets. The integration of online tourist data is conducive to the quantitative evaluation or comparison of world cultural and natural heritage tourism resources, and to find out the areas that need to be upgraded and improved in terms of tourism of World heritage sites.

2.The author(s) need to think about the implication of this paper. Can the results of this paper present some new insight to the scholars or can the results give some practical suggestions to the government?

Thank you very much. By collecting the user data of China's three major OTA platforms, this paper constructs the attractiveness evaluation system of China's world cultural and natural heritage tourism resources, aiming to explore the attractiveness level of world cultural and natural heritage from different dimensions, and explore the attractiveness characteristics of tourism resources from space. There are two main innovations:

1. Data innovation. The data in this paper come from the real ratings and comments of tourists in different aspects of the three well-known tourism platforms in China, which ensures the authenticity, reliability and availability of the data. The data source is not limited by region, and the real views of tourists can be understood more comprehensively.

2. Method innovation. According to the national standard GB/T18972-2017 "Classification, Investigation and Evaluation of Tourism Resources" issued by China, this paper divides China's world cultural and natural heritage tourism resources into five main categories according to the specific causes and status of tourism resources: Geographical landscape, aquatic landscape, biological landscape, architecture and facilities, and historical sites are subdivided into ten sub-categories. Using the kernel density and the Moran index from a finer dimension, the different attractiveness of different resource types is compared in geographical space.

The innovation of the above research content can provide a new research perspective for scholars. This study reflects the attractiveness of tourism resources of China's world cultural and natural heritage in different dimensions. In practice, it can also provide guidance and suggestions for scenic spots with different resource types to strengthen tourism attraction construction according to their disadvantage dimensions.

3.For the kernel density analysis, because I did not obtain the copyright of the map, I deleted the previous drawing, please understand, thank you.

---

## [Editor Report · Decision Letter 1]

11 Jul 2023

Evaluation of attraction and spatial pattern analysis of world cultural and natural heritage tourism resources in China

PONE-D-23-13162R1

Dear Dr. Long,

We’re pleased to inform you that your manuscript has been judged scientifically suitable for publication and will be formally accepted for publication once it meets all outstanding technical requirements.

Kind regards,

Qingsong He, Ph.D.

Academic Editor

PLOS ONE

---

## [Editor Report · Acceptance letter]

27 Jul 2023

PONE-D-23-13162R1 

*Evaluation of attraction and spatial pattern analysis of world cultural and natural heritage tourism resources in China*

Dear Dr. Long:

I'm pleased to inform you that your manuscript has been deemed suitable for publication in PLOS ONE. Congratulations! Your manuscript is now with our production department. 

Kind regards, 

on behalf of

Dr. Qingsong He 

Academic Editor

PLOS ONE